# Hierarchical dinucleotide distribution in genome along evolution and its effect on chromatin packing

Zhicheng Cai[1,2,3], Yueying He[1,2,3], Sirui Liu[1,2,3], Yue Xue[1,2,3], Hui Quan[1,2,3], Ling Zhang[1,2] , Yi Qin Gao[1,2,3]

**Dinucleotide densities and their distribution patterns vary significantly among species. Previous studies revealed that CpG is susceptible to methylation, enriched at topologically associating domain boundaries and its distribution along the genome correlates with chromatin compartmentalization. However, the multiscale organizations of CpG in the linear genome, their role in chromatin organization, and how they change along the evolution are only partially understood. By comparing the CpG distribution at different genomic length scales, we quantify the difference between the CpG distributions of different species and evaluate how the hierarchical uneven CpG distribution appears in evolution. The clustering of species based on the CpG distribution is consistent with the phylogenetic tree. Interestingly, we found the CpG distribution and chromatin structure to be correlated in many different length scales, especially for mammals and avians, consistent with the mosaic CpG distribution in the genomes of these species.**

## Introduction

The genome composition and organization play important roles in various cellular processes. To adapt to the environment, the genomes of organisms undergo drastic mutations, leading to developmental complexity in organisms (Suzuki & Nijhout, 2006; Liu et al, 2014). Many studies were conducted to explore the relationship between genome organizations and diverse phenotypes of organisms (Kim et al, 2004; Pigliucci, 2010; Kulminski et al, 2013; Rinker et al, 2019).

Owing to the development of next-generation sequencing, the genomes of a large number of species are now available. They are different in sizes, karyotypes and base compositions. Particularly, it has been long known that the proportion of nucleotides varies significantly among species. For example, there is a substantial variation in average G+C contents among different species. In prokaryotes, G+C contents were reported to be positively correlated to the optimal growing temperature (Musto et al, 2004). Accordingly,

thermal stability of DNA double helix was reported to be influenced by G+C contents (Yakovchuk et al, 2006), although thermophilic archaea with extreme low G+C contents also exist. Besides, the nucleotide composition also influences gene functions and regulation. The GC-rich genes in grass are usually related to basic metabolic processes and biotic stress responses (Tatarinova et al, 2010). In yeast, the AT-rich sequences are ubiquitous in promoter regions and incorporate poorly into nucleosomes, and are thus important for transcription initiation. Besides, human promoters are divided into two classes according to the CpG density (Saxonov et al, 2006). Genes with high CpG density promoters are generally expressed in more tissues than those with low CpG density promoters.

Moreover, the nucleotides and dinucleotides are not uniformly distributed along the genome. The proportions of C and G vary along the chromosomes over a large genomic length scale. The chromosomes of warm-blooded vertebrates are divided into isochores (Bernardi et al, 1985; Bernardi, 1993), which are different DNA segments with homogeneous G+C content and are separated by the sharp content transition. Isochores correlate with genome features such as gene density and replication timing (Costantini et al, 2006; Costantini & Musto, 2017). Among dinucleotides, CpG distribution is the most thoroughly studied, for its biological significance. CpG is deficient in the genomes of vertebrates, probably because DNA methylation occurs predominantly at CpG. Besides, CpG is enriched at many human promoters, and high gene densities are often found in the CpG-rich regions in the human genome. Accordingly, CpG has a higher tendency to be unevenly and hierarchically distributed, that is, CpG aggregates to form the CpG islands (CGIs), the distribution of which is also heterogeneous and correlated with gene density on the genome. Based on the distribution of CGIs, the human and mouse genomes can be divided into two types of Mb-sized domains: CGI (gene) forest domains with high CGI (gene) density and CGI (gene) prairie domains with low CGI (gene) density (Liu et al, 2018). Consistent with such multi-scale uneven CpG distributions, long-range correlations have been found to exist in the distributions of nucleotides and dinucleotides by methods such as power spectrum (Voss, 1992; Buldyrev, 2006), detrended fluctuation analysis (Peng et al, 1992), and wavelet transform (Audit et al, 2001;

[1]Beijing National Laboratory for Molecular Sciences, College of Chemistry and Molecular Engineering, Peking University, Beijing, China  [2]Biomedical Pioneering Innovation Center, Peking University, Beijing, China  [3]Beijing Advanced Innovation Center for Genomics, Peking University, Beijing, China

Correspondence: gaoyq@pku.edu.cn

 **Life Science Alliance**

Arneodo et al, 2011). However, studies are yet to be expanded beyond limited number of model species and to cover the entire genomes but not limited DNA regions such as exons and introns. To find out how multi-scale CpG distributions change along evolution and its connection to gene regulation in different species, a comprehensive study of CpG distributions in the genomes covering species from different taxa is strongly desired.

It becomes increasingly accepted that the chromatin 3D structure plays an important role in gene regulation and cellular functions (Lieberman-Aiden et al, 2009; Bickmore & van Steensel, 2013). In the meantime, many factors contributing to the chromatin structure formation have been explored. For example, it was suggested that compartments are fine-scale structures of chromatin which are correlated with transcription states (Rowley et al, 2017). Topologically associating domains (TADs) are highly conserved structures among cell types in mammals, the boundaries of which often correspond to CTCF loop anchors (Dixon et al, 2012; Rowley & Corces, 2016). Recent studies have revealed the influence of linear DNA sequence on 3D chromatin organizations. TAD boundaries often correlate with the presence of CGI, and the CGI distribution along the genome is correlated with compartmentalization in human and mouse cells (Liu et al, 2018). A unified and quantitative analysis on how the linear CpG distribution affects the 3D chromatin organizations at various length scales is expected to shed more light on the mechanism of chromatin structure formation as revealed by Hi-C and imaging studies.

A sequence-based model (Liu et al, 2018) was proposed to explain the chromatin structure formation from the domain segregation perspective, which provides a framework for the exploration of chromatin structure formation, especially compartmentalization, in various cellular processes. To verify and generalize this model, different species besides human needs to be examined. It is also interesting to interrogate how this sequence–structure relation affects the different phenotypes of species.

In this study, we use quantitative methods to analyze the multi-scale CpG distributions on the genomes of a number of species from different taxa. We showed that using the disparity of CpG local density fluctuation one can effectively cluster species into different groups, consistent with their positions on the phylogenetic tree. The distribution of CpG is also characterized by the multi-scale entropy (SE) and the Pearson correlation in a scale-continuous manner. More importantly, we quantified the relation between CpG distribution, 3D chromatin organizations, and gene expression activity for a number of exemplary species. Our results also show that the CpG distribution profile correlates well with the degree of a species' chromatin structural segregation and body temperature control.

## Results

CpG has many special properties among the 16 dinucleotides, such as its often low density in the genome (Cooper & Gerber-Huber, 1985) and richness in many human promoters in the form of CpG islands (CGIs) (Saxonov et al, 2006). We therefore first focused on the CpG dinucleotide because of these known relation to biological functions. Next, we extended our analysis to all dinucleotides.

### Distribution of CpG along the genomes of different species

We compared the CpG density heterogeneity of different species based on the amplitude variation of its density fluctuation along the DNA sequence. Here the CpG density was averaged using a 1,000-bp window which is close to the average length of CGI. Next, we decomposed CpG density data series of different species along the DNA sequence into fluctuations at different frequencies by Hilbert-Huang transform (HHT) (see the Materials and Methods section and Supplemental Data 1). We then defined and calculated the variability (see the Materials and Methods section) of the CpG density fluctuation at the highest frequency, a large value of which corresponds to a large amplitude difference in CpG density fluctuation along the DNA sequence.

As shown in Figs 1 and S4, the CpG variability can be used to effectively cluster species into different groups. Among the species investigated, birds possess the highest variability, and bacteria have the lowest. The variabilities of mammals, reptiles, fishes, plants, and invertebrates are intermediate. Such an order approximately follows the phylogenetic tree. From the ranking of variability, alligator is close to mammals, and is thus closer to those of birds than to other reptiles (especially, lizards). Interestingly, according to the phylogenetic tree, birds did evolve from reptiles and alligators are more closely related to birds than to other reptiles (Crawford et al, 2012). The ranking of reptile variabilities also fits to their positions in the phylogenetic tree. In addition, platypuses have the lowest variability among mammals, again consistent with its position in the phylogenetic tree.

As a different measure, we also analyzed the multi-SE and Pearson correlation for CpG density of the genomes of different species to compare their CpG distribution in a scale-continuous manner. Consistently, mammals and birds have larger multi-SE than bacteria, plants, invertebrates, and fishes at large genome length scales (Fig S2A–H). Besides, Pearson correlation of the CpG density of mammals and birds decay with the increase in genomic distance roughly in a form of power law (Fig S3A–D), whereas in bacteria, plants, invertebrates, and fishes this power law decay only persists to a short distance if it does exist. These results all indicate a more heterogeneous CpG distribution in mammals and birds than in bacteria, plants, invertebrates and fishes.

### Division of CGI-rich and CGI-poor domains in different species

In our previous study, CGI forest and prairie domains were defined in mouse and human based on the unevenness of CGI distribution along the DNA sequence (Liu et al, 2018). These two sequential domains effectively reflect the linear segregation in the genome of not only CGI densities but also genetic, epigenetic, and structural properties. In this study, we generalize the definition of CGI forest and prairie domains to all species termed CGI-rich and CGI-poor domains.

Because for genomes of most species traditionally, defined CGIs cannot be identified because of high CpG density or a largely even distribution of CpG, a method is needed to generalize the CGI forest-prairie domain definition (see the Materials and Methods section). We make use of properties of the prairie domains of human and mouse to define the "CGI-poor clusters," and the generalized CGI-rich

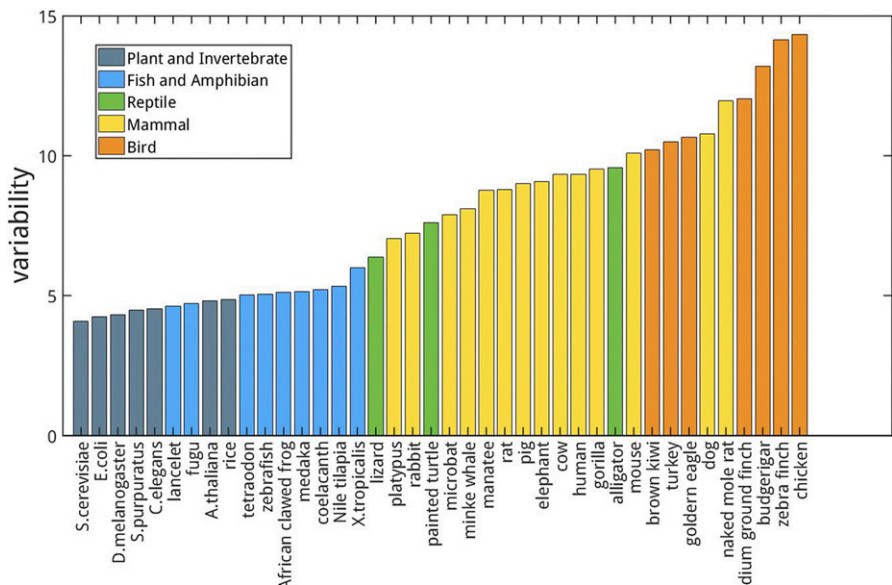

**Figure 1.  Variabilities of the CpG density of the longest chromosome for different species.**

and CGI-poor domains are defined so that CGI-rich domains are the longest possible domains that possess little "CGI-poor clusters." The generalized CGI-rich domains are regions in which high CpG density loci cluster, whereas the generalized CGI-poor domains have low CpG density and are deficient in CGIs. The newly defined domains are in good accordance with previous definition (the CGI-rich domains overlap ratio with the CGI forest domains is 94% for human and 87% for mouse). For representative species, we evaluated their sequence properties, including the proportion of CGI-poor domain, CGI-poor domain average length, average CpG densities of the two types of domains, and the corresponding coefficients of variation (CVs) of the CpG density.

Among the species analyzed, *Escherichia coli* has very few CGI-poor domains with very low average CGI-poor domain lengths. The CpG densities of *E. coli* are significantly higher than multicellular eukaryotes. The *E. coli* genome is gene-rich and lacks noncoding elements, which is consistent with the fact that it contains very few CGI-poor domains. For eukaryotes, fishes, and amphibians also have low amounts and short lengths of CGI-poor domains, but their CpG densities in both CGI-poor and CGI-rich domains are lower than the corresponding domains in prokaryotes and invertebrates. Reptiles possess more CGI-poor domains and the CpG density levels of their CGI-poor domains are lower than fishes and amphibians, close to birds and mammals. In fact, birds and mammals are significantly different from other species, with a higher proportion of core CGI-poor domains (0.229 ± 0.06 compared to 0.003 ± 0.007) and a longer average CGI-poor domain length (1.76 ± 0.63 Mb compared to 0.23 ± 0.21 Mb), indicating an uneven CpG density distribution at the Mb level. The density fluctuation of the low CpG density regions is small, indicating that these regions have largely uniformly distributed CpG dinucleotide (with an average standard deviation of 27.6 ± 4.3 Mb$^{-1}$ for mammals and birds, 55.0 ± 86.4 Mb$^{-1}$ for other species). Birds and mammals also differ from each other in sequential properties. Mammals have longer average CGI-poor domain lengths (2.03 ± 0.57 Mb) than birds (1.25 ± 0.36 Mb). The

average CpG densities for CGI-rich domains of mammals are slightly higher than those of birds, and their CGI-rich domain CpG densities vary significantly less.

More generally, hierarchical clustering yielded a dendrogram for different species (Fig 2), which is in reasonable accordance with the phylogenetic tree. It can be seen from Fig 2 that *E. coli* can be distinguished from eukaryotes. Among eukaryotes, plants specifically cluster together. Moreover, fishes and amphibians are also distinctly discriminated against birds and mammals. Birds and mammals roughly separate from each other except for turkey and brown kiwi of which the CpG density fluctuations at the 1-kb average length in CGI-rich domains are also similar to those of mammals. Interestingly, lizard and painted turtle are grouped closely with coelacanth, which is an important link in vertebrate evolution from fishes to tetrapod.

### Average CpG density and CpG distribution in different species

As seen in Fig 3B, the scatterplot of average CpG density and CpG variability of the different species shows roughly an "L" shape. In general, along evolution, the CpG density decreases. For example, its value for invertebrates and plants is in general much higher than that for fishes, reptiles, birds, and mammals. The CpG density difference among different invertebrates, plants, and that between invertebrates and vertebrates are all very large. Such a difference is small among different species of vertebrates: the average CpG density of mammals and birds is only slightly lower than that of fishes and reptiles. The average CpG density being nearly constant during the evolution of mammals and birds is consistent with a detailed balance condition in CpG mutation, that is,

$$u \times f = v \times (1 - f).$$

In the above equation, $u$ is the mutation rate of CpG, $f$ is the density, and $v$ is the reproduction rate of CpG. Besides mutation,

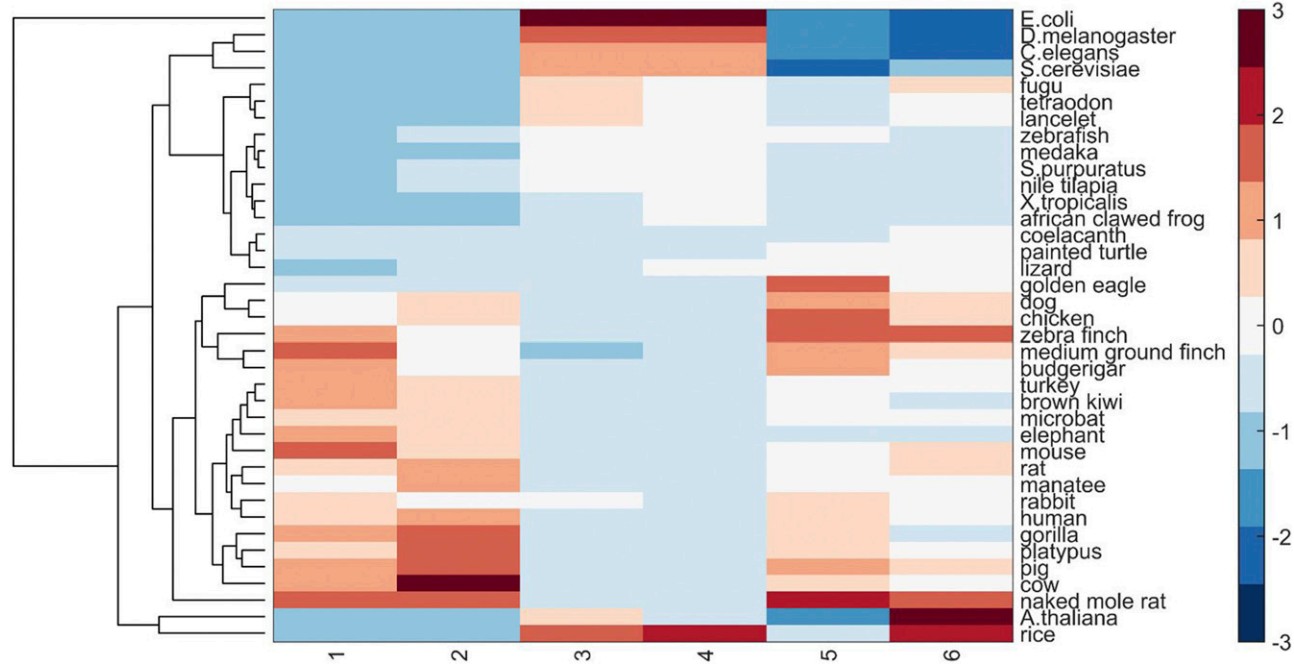

**Figure 2.  Dendrogram for 38 species from different classes.**
Six columns represent the proportions of CGI-poor domain, CGI-poor domain average length, average CpG densities of CGI-rich and poor domains, and the corresponding coefficients of variation (CVs) of CpG density, respectively. The color bar shows the column-scaled values.

insertion of repeating sequences including CpG-rich ones into their genomes (partly from the transposition [Hwu et al, 1986]) could also be important in avoiding further CpG depletion in genomes and could explain the nearly constant average CpG density among mammals and birds.

Different from CpG density, the CpG variabilities of invertebrates and plants are all generally very similar and small. In contrast, the variabilities of mammals and birds are much higher than those of other species (Fig 3B). Therefore, the genomes appear to evolve in different ways among different species. During the evolution of invertebrates and plants, the CpG density decreases with little change of the genome mosaicity and during the evolution of mammals and birds, the CpG density remains largely constant but the genome mosaicity increases. As a comparison, C+G content does not exhibit a clear trend of changes among species from different classes like CpG density (Fig 3A).

### CpG distribution and 3D chromatin organizations

Given that 3D structure is typically strongly influenced by 1D sequence (such as that seen in protein folding), the DNA sequence appears to be of dramatically different properties for different species. In this section, we investigate how the CpG distribution correlates with the 3D chromatin organizations in different species. We collected the Hi-C datasets of archaea, yeast, *Arabidopsis thaliana*, zebrafish, chicken, mouse, and human. To compare the two-dimensional Hi-C map with the CpG distribution along the linear genome, we factorized the Hi-C matrix into two one-dimensional vectors by nonnegative matrix factorization (NMF, see the Materials and Methods section). We termed these two

vectors "structure vectors" (i.e., W1 and W2), which reflect intensities of two anti-correlated Hi-C signals along the sequence. We tested the robustness of structure vectors and found structure vectors factorized by different NMF methods to be similar to each other after scaling (Fig S6A–F).

Next, we performed continuous wavelet transformation for the structure vectors as well as for the CpG density along the sequence at multiple wavelet frequencies, to compare directly their fluctuations at varied length scales. Interestingly, the wavelet transform coefficients of structure vectors and CpG density of the same species largely resemble each other at many length scales (Fig 4A and B and S7A–D). To quantify their similarity, we further calculated the Pearson correlation of the wavelet transform coefficients of structure vectors and CpG density. One structure vector is overall positively correlated, whereas the other is anti-correlated with CpG density (Figs 5A and B and S8), indicating that the DNA can be grouped into high and low CpG density groups in space and contacts tend to be formed within each of the two groups. In birds and mammals such as chicken, mouse, and human, the correlation increases with the length scale, and the peak appears at the ~10 Mb scale. In contrast, this correlation in other species such as *A. thaliana* and zebrafish does not increase with the length scale monotonously. Consistently, we also compared the correlation between CpG-rich/poor domains and compartments A/B in different species and found that it is higher in human, mouse and chicken than in *A. thaliana* and zebrafish (Table S9 and Fig S9A and B). Such a result suggests that the correlation between CpG distribution and 3D chromatin organization increases along evolution, especially in large length scales, indicating the strengthened influence of globally CpG heterogeneous distribution on chromatin

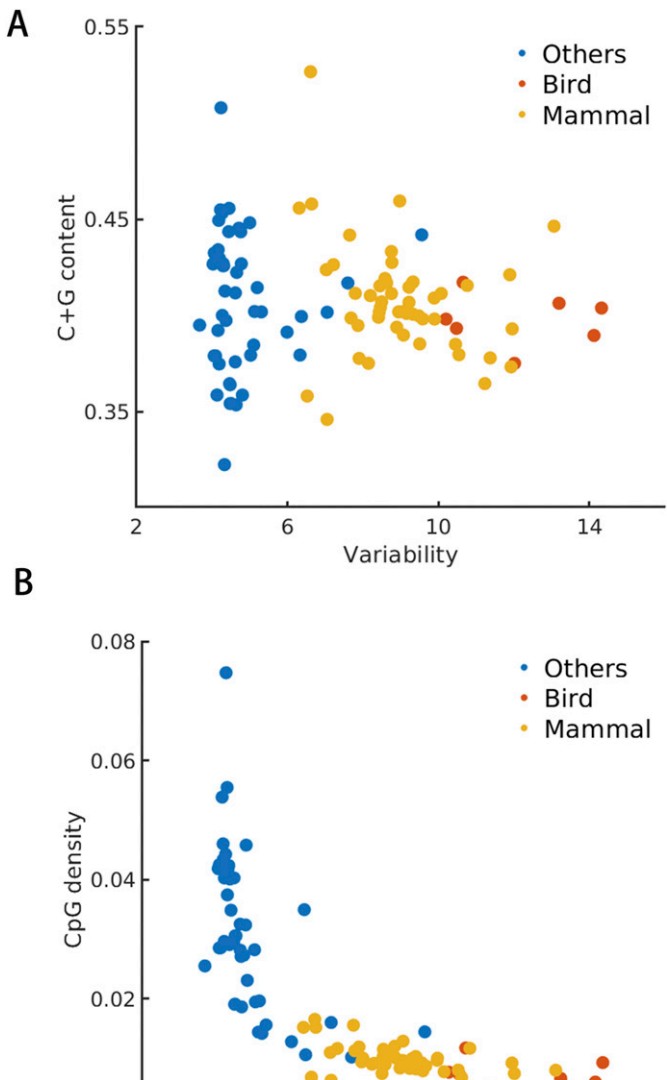

**Figure 3.** **(A)** The scatterplot for C+G content and CpG variability of various species. **(B)** The scatterplot for CpG density and CpG variability of various species.

structure in birds and mammals than other species such as fishes and plants, consistent with the increasing multi-scale heterogeneity of CpG distribution (Figs 1 and S2A–H).

Then, we compared CpG distribution with the lengths of TADs and compartments in human (Fig 5C). At length scale of compartments, CpG density is highly correlated with the structure vectors in IMR90 cell line, whereas their correlation is weaker at the length scale corresponding to TADs. Consistently, the overlap between A/B compartments and CpG-rich/poor domains is more significant than the one between TADs and CpG-rich/poor domains (Fig S10). These observations suggest CpG density may play a more important role in compartment than in TADs formation.

Because CpG distribution and 3D chromatin organization is highly correlated in species with uneven CpG distribution, one may speculate the chromatin structure to be more segregated in mammals and birds. To examine this hypothesis, we calculated the chromatin contact frequency decay for different species. One can see from Fig S11A and B that contact frequency decay exponents are very similar between mouse and human, and among different cell types of human. At the short length scale, the Hi-C contact frequency for mouse and human samples decays more slowly than species such as *A. thaliana*, yeast, and drosophila, indicating a more segregated chromatin structure of mouse and human at these length scales, consistent with the formation of interactions within the linear DNA domains such as compartments (Liu et al, 2018). In contrast, at the length scale larger than ~Mb, the chromatin contact frequency for *A. thaliana*, yeast and drosophila decays more slowly, indicating that long-range contacts are more likely to be observed in these species than in human and mouse. Besides, we also calculated the interaction segregation ratio of compartments A/B for different species (Table S10). The interaction segregation ratio of human and mouse is higher than other species.

## CpG distribution and gene expression level

It was previously reported that the gene expression level is higher in CpG-rich than in CpG-poor regions (Liu et al, 2018). It is interesting to examine at which scale(s) such a correlation between CpG density and the gene expression hold. We found that a reasonably strong correlation exists at the scales ranging from hundreds of thousands to millions of base pairs between gene expression and CpG distribution in species such as human and zebrafish which have an uneven CpG distribution (Figs 6A and C and S12). However, in species with largely even CpG distribution like plants, the gene expression level varies along the sequence and is only weakly correlated with the CpG distribution (Fig 6B and C). Furthermore, we grouped genes by CpG density and compared the expression levels of genes of different CpG densities. The dependence of gene expression level on CpG density is seen to be different for different species (Fig S13A–C). For example, the expression levels of zebrafish and rice genes show a more pronounced peak at median CpG densities than that of human. These results suggest that the relation between CpG density and expression level becomes stronger in evolution, with their distribution along the genome becomes more heterogeneous.

## Distribution properties of other dinucleotides

Finally, we extended our analysis to all 16 dinucleotides. Overall, we found that CpG tends to possess the highest density distribution fluctuation among the 16 different dinucleotides. For example, CpG density has the largest fourth moment at various length scales (Fig 7A), indicating its highest probability of assuming extreme values among all the 16 dinucleotides. We also calculated the variability of all dinucleotide densities. The result given in Fig 7B again shows that CpG has the largest local fluctuations.

We also found that dinucleotides composed by C and G (i.e., CpC, GpG, CpG, and GpC) or by A and T (i.e. ApA, TpT, ApT, and TpA) have a fourth moment that is larger than the other eight dinucleotides such as ApG which is composed of two types of nucleotides, one being A or T and the other being C or G (Fig 7). Besides CpG,

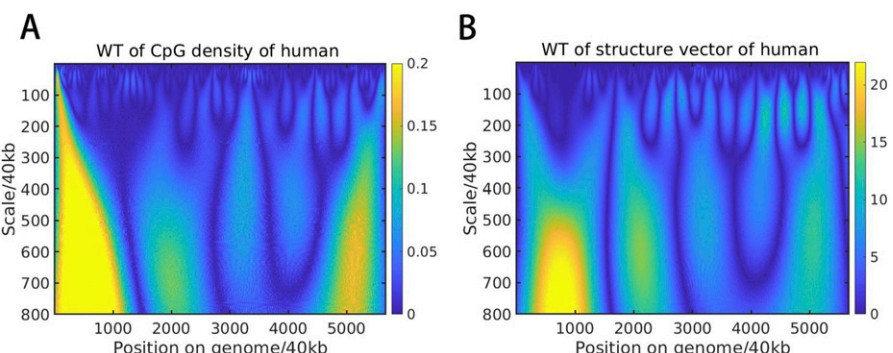

**Figure 4.** **(A, B)** Wavelet transform of (A) CpG density and (B) structure vector along the chr1 of human liver cell at different length scales.

distributions of other dinucleotides also become more heterogeneous changing from bacteria, invertebrates, plants, fishes to mammals and birds. For example, the Pearson correlations of CpA densities of mammals and birds but not bacteria, invertebrates, plants, and fishes exhibit a long-range power law decay (Fig S3E and F). Interestingly and consistent with the trend discussed above, among all the mammals, the density correlation of platypus decays more quickly than other mammals. The exponent of the decay

**Figure 5.** **(A, B)** Pearson correlation of wavelet transform coefficient of structure vectors (i.e., W1 and W2) and CpG density of different species compared with random. X-axis is the length scale of wavelets. **(C)** Top: the length distribution of compartments and topologically associating domains of human, respectively, Bottom: Pearson correlation of wavelet transform coefficient of structure vectors and CpG density of chr1 of IMR90 cell line.

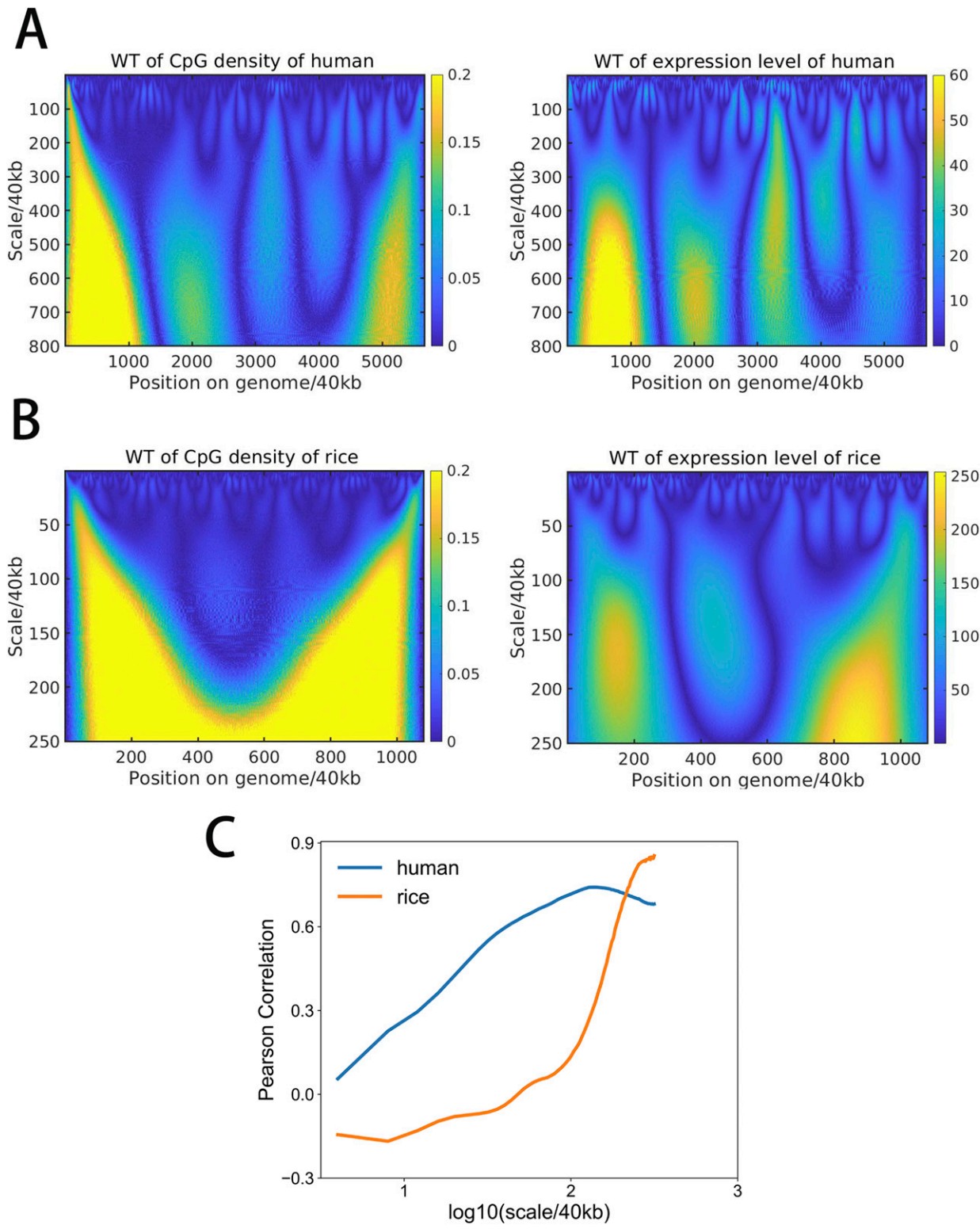

Figure 6. **(A)** Wavelet transform of CpG density and expression level averaged at 40 kb along chr1 of human liver cell. **(B)** Wavelet transform of CpG density and expression level averaged at 40 kb along chr1 of rice. **(C)** Pearson correlation of wavelet transform coefficient of CpG density and expression level at multi-length scales of human and rice.

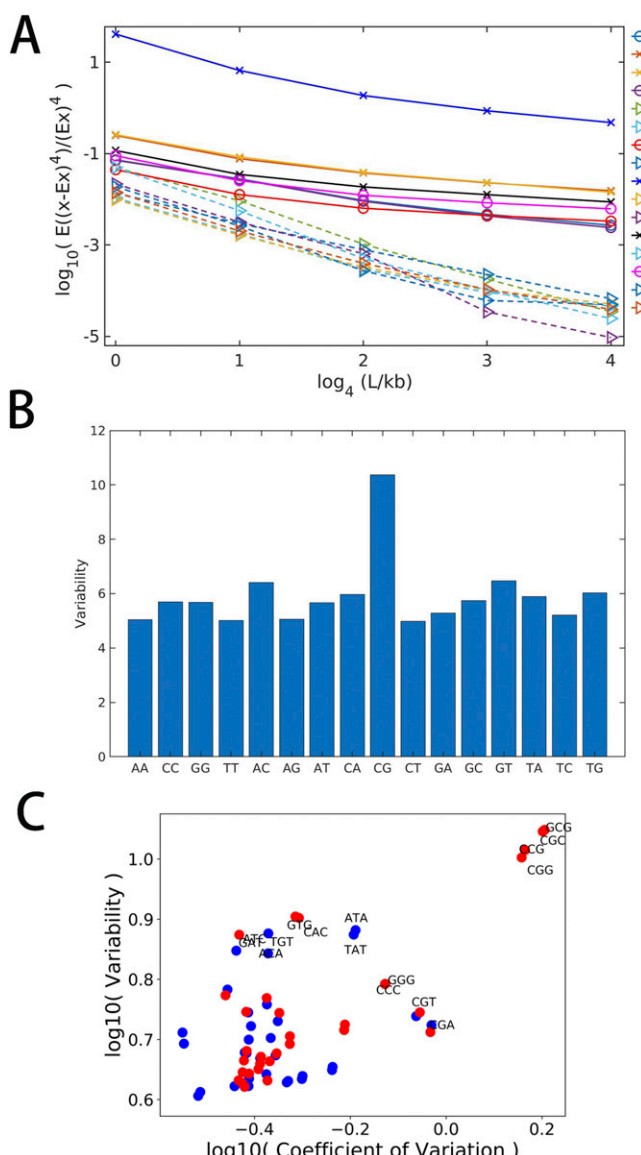

**Figure 7. Distribution characteristics of dinucleotides and trinucleotides of human chr1.**
**(A)** Fourth moment of 16 dinucleotides density averaged at different length scales. **(B)** Variability of 16 dinucleotides density averaged at 1 kb. **(C)** The scatterplot for variability and coefficient of variation of 64 trinucleotides. Blue: trinucleotides beginning with A or T; Red: trinucleotides beginning with C or G. Only trinucleotides with large variability or Coefficient of Variation are labeled.

function for platypus is similar to that of fishes, possibly because of its transitional position on the phylogenetic tree.

We further analyzed the density fluctuation of longer DNA sequences, for example, trinucleotides and tetranucleotides. The density fluctuation of these longer sequences appears to be largely determined by that of dinucleotides. For example, as shown in Fig 7C, the trinucleotides containing CpG such as CCG, CGC and CGA also show high Coefficients of Variation. Meanwhile, the trinucleotides such as ACA have a high variability similar to those of AC and CA (Fig 7B and C). Similar phenomena are also observed for tetranucleotides (Fig S14A–D). Interestingly, it is known that codon is

dominantly determined by its first dinucleotide (Hartman, 1975), consistent with the observed importance of the dinucleotide composition in the DNA sequence.

## Discussion

On one hand, chromatin structure formation and gene transcription regulation form complex networks and are influenced by many genetic and epigenetic factors. On the other hand, simple sequence information such as nucleotide and dinucleotide density distribution appears to have strong predictive power on these various biological functions. For example, CpG dinucleotide has been shown to influence gene regulation and chromatin organization at different levels, ranging from single CpG through the methylation to CGI in promoter regions and TADs boundaries. CGI distribution along the genome was also shown to correlate with compartmentalization globally in human cells. These results show that the multi-level heterogeneous CpG (among other dinucleotides) density distribution does code important information on chromatin organization and gene expression.

The sequence properties of bacteria, plants, invertebrates, fishes, and reptiles are distinctly different from birds and mammals in terms of the dinucleotide distribution. The CpG density of the former shows low variabilities and a decreasing multi-SE as a function of genomic distance. Consistently, its Pearson correlation coefficient does not exhibit a long-range power law decay. In contrast, the CpG density of birds and mammals has relatively high CpG variabilities, a multi-SE generally increasing with the genomic length, and a wide-range power law decay, all of which indicate a more heterogeneous CpG distribution. From the perspective of CGI-rich and CGI-poor domains, birds and mammals have a larger proportion of and longer CGI-poor domains than other species. Consistently, birds' and mammals' overall average CpG density is also lower than that of other species.

Interestingly, species with shorter and smaller proportions of CGI-poor domains are more dominantly cold-blooded, living with a wide body temperature range. CGI-poor domains rarely exist in genomes of these species, but become ubiquitous in birds and mammals. Birds' and mammals' long CGI-poor domain lengths, relatively high proportions of CGI-poor domains, and significant CpG density differences between these two types of domains suggest a high sequence heterogeneity. These two species are normally warm-blooded living with a narrow body temperature range.

Consistently, the values of CpG density variability also show a correlation with body temperature control of different species (Fig 1). Warm-blooded species (birds and mammals) have higher DNA sequence variabilities than cold-blooded ones (fishes, amphibians, and reptiles). Among warm-blooded species, DNA sequences of birds tend to have a higher variability than those of mammals. Furthermore, among the cold-blooded species, alligators are known as "half warm-blooded animals" because of their maintenance of relative high body temperatures through basking (Seebacher, 2003; Tattersall et al, 2012). Consistently, the CpG density variability of the alligators resembles that of mammals and

is very different from either turtles or lizards. To further investigate the possible relation between the body (or environmental, in the case of cold-blooded species) temperature and the CpG density variability, we also calculated and compared the variability of fishes from the tropical and polar regions (Table S4). The tropical fishes were found to have a higher sequence variability than the polar fishes. Interestingly, the brown bear appears to have a higher sequence variability than the polar bear (Table S3 and Fig S15A and B). Such a coincidence may suggest a possible relation between the living environment and the CpG distribution of the genomes of different species.

As a note, the DNA methylation level was also reported to be negatively correlated with the environment temperature for fishes and reptiles (Varriale and Bernardi, 2006a, 2006b). It is well known that the most of the methylation occurs on cytosine and higher CpG density often results in the higher methylation level. Such an observation is consistent with our results for a high CpG density is correlated with the low variability (Fig 3). In fact, earlier analysis shows that the methylation level correlates with CpG density and its fluctuation (Liu et al, 2018).

As mentioned above, a close relation exists between the mosaic CGI and gene distributions in human and mouse genomes (Liu et al, 2018). The segregation of the CGI/gene-rich and CGI/gene-poor domains largely determines the chromatin compartmentalization, with a large number of alternating compartments A and B (Fig S16A). In contrast, it is well known that chromosomes of many plants are simply partitioned into three compartments: two compartments A near the telomere and one compartment B near the centromere (Liu et al, 2017). We calculated the gene density of rice and *A. thaliana* along the genome and found that their gene density correlates well with compartment partition and the gene density is higher in compartment A than in B (Fig S16B and C). However, little correlation is seen between CpG density and compartment partition for these species as the CpG distribution is nearly uniform in plants (Figs 1 and S2). Similar gene density–compartment correlation is also observed for archaea (Fig S16D). These results suggest an important role of the gene distribution along the linear genome in chromatin structure segregation and compartment formation, and suggest that the uneven CpG distribution further promoting the spatial partition of the chromosome in species such as mammals and birds but not plants. The separation of the chromatin structure into different compartments following the gene density distribution is highly conserved across different species, and indicates a connection between the CGI (and thus CpG, as well as other dinucleotides) distributions and the gene clustering along the linear genome. In fact, genes of similar functions also tend to segregate along the linear genome (Lewis, 1992, 2004), consistent with a function-driven gene and DNA sequence redistribution in evolution. Because genes of similar functions tend to form spatial contacts (Belcastro et al, 2011; Ibn-Salem et al, 2017), it would be interesting to examine the cross-talk between 3D chromatin structure and DNA linear sequence in different species and in evolution.

Recently, it was proposed that a stable phase separation of genome is correlated to differentiation, senescence, and diseases such as cancer (Liu et al, 2018). A high mosaicity of a genome appears to correspond to stable differentiation. Consistent with this

theory, with a low genome mosaicity and a largely even CpG distribution (see the Discussion section above and Fig 1), plants are prone to reprogramming and dedifferentiate. For example, plants can generate calluses in response to stresses, many of which are totipotent (Steward et al, 1958; Ikeuchi et al, 2013). Moreover, fishes and reptiles can remain growing and developing throughout their lives. For example, lizards can regenerate tails (Baranowitz et al, 1979). Among mammals, whales, which have a low CpG variability (Fig 1), are the largest mammals on the earth and famous for the longevity and ability of suppressing cancer (Caulin & Maley, 2011). Similarly, elephants are also known for their longevity, resistance to aging and cancer, and indeterminate body size. In contrast, cells of birds and mammals with a high sequence mosaicity are difficult to reprogram (Surani, 2012), and more prone to cancer than invertebrates and plants (Albuquerque, 2018).

Finally, our analysis indicates that CpG density, consistent with its highly uneven distribution, is a better function and structure indicator than C+G content. CpG density but not C+G content (Figs 1 and 3) can cluster species corresponding to their positions in the phylogenetic tree. It was also found that the CGI-rich and CGI-poor domains correlate more strongly than isochores to the segregation of the genomic features such as compartment and TADs formation as well as DNA methylation in human and mouse (Liu et al, 2018), indicating CpG density correlates better than the C+G content to the chromatin structure formation. Because C to T mutation is believed to be associated with CpG methylation, which has been shown to be actively involved in gene regulation, these results all suggest that the CpG density is likely more directly connected to biological functions and evolution than C+G content.

## Conclusion

In this study, we explored the nucleotide distribution features in the genome sequences of different species. In evolution, the genome gradually loses CpG dinucleotide and gains in the unevenness of their distributions along the genome. Among the dinucleotides, the density distribution of CpG shows the most prominent multi-scale heterogeneity. Based on this distribution, we divided the genomes into the CGI-rich and CGI-poor domains with distinct compositions and properties. By analyzing the average lengths, ratios and compositions of the CGI-rich and CGI-poor domains, we showed that genomes of birds and mammals are more heterogeneous than those of bacteria, plants, invertebrates, fishes and reptiles. Furthermore, we found that the CpG distribution is closely correlated with 3D chromatin organization at different length scales, especially in birds and mammals, suggesting the increased role of DNA sequence in determining the chromatin 3D structure in evolution. In warm-blooded animals, it appears that chromatin organization is strongly coded in the DNA sequence, especially in the uneven dinucleotide distributions. Because of the genome mosaicity, the chromatin 3D structures of warm-blooded species are likely to be more (stably) segregated than the ones of cold-blooded species. The increased role of DNA sequence in determining the chromatin structure along evolution is consistent with a co-evolution of the 1D and 3D genomes. We speculate this difference in genome sequence segregation to have an effect on differentiation, senescence, and maybe susceptibility to cancer among species. The various

correlations found in this study call for careful and extensive experiment verifications.

# Materials and Methods

### Data source

In this study, we analyzed the genomes of 38 representative species including bacteria, plants, invertebrates, fishes, reptiles, mammals, and birds (Table S1). Genomes of species were retrieved from UCSC genome browser and NCBI. For each species, the longest chromosome was chosen to calculate the variability, multi-SE, and Pearson correlation coefficient of the dinucleotide density. Source and information of Hi-C datasets are listed in Table S5. Source of RNA-seq are listed in Table S6. Cell types of Hi-C map are listed in Table S7.

### HHT

A brief explanation of HHT (Huang et al, 1998) is given here and the details can be found in the reference and Supplemental Data 1. HHT can decompose a data series into oscillatory modes of different frequencies. The difference between HHT and Fourier transform is that HHT can be applied to nonlinear and nonstationary series without the requirement of a priori basis.

For a given data series X(t) as input, the mean value series $m_1$ of the upper and lower envelope of X(t) is calculated using the cubic spline lines. The difference $h_1$ between the input X(t) and $m_1$ is called the first protomode,

$$h_1 = X(t) - m_1.$$

Here, $h_1$ is used as input in the next iteration to yield a new set of $h_1$ until the following conditions are satisfied: (1) In the entire dataset, the number of extrema and the number of zero crossings must either be equal or differ at most by one. (2) At any data point, the mean value of the envelope consisting of the local maxima and the one consisting of the local minima is zero.

After the first mode $h_1$ has converged, the difference between the input X(t) and $h_1$, that is, X(t) – $h_1$ can be used as the input for the next iteration to obtain the next mode $h_2$. Repeating this procedure, we can retrieve modes of different frequencies: $h_1$, $h_2$, $h_3$...$h_n$, in the order of the decreasing frequency.

Take the CpG density series of human (Fig S1A and B) and Nile tilapia (Fig S1C and D) as example, they can be decomposed to series at different frequencies by HHT.

### Dinucleotide variability

We define the variability of decomposed dinucleotide density series to quantify the extent of the dinucleotide density fluctuations at high frequencies. Given that the decomposed series fluctuates around the value zero (Fig S1), we calculated the absolute values of the decomposed series and then divided the new (absolute value) series into the high amplitude group in which the values are larger than $a + 3\sigma$, and the low amplitude group with values smaller than

$a + \sigma$. Here a and $\sigma$ are average and standard deviation of the new series. CpG variability is defined as follows:

$$V = \frac{a_{high}}{a_{low}},$$

where $a_{high}$ and $a_{low}$ are high and low amplitude group averages, respectively (The threshold value $a + 3\sigma$ is chosen because $3\sigma$ is often used as a threshold in detecting outliers in statistics theory. Choice of different thresholds does not have significant influence on the relative order of CpG variability of different genomes, see Table S2). A higher variability for a genome indicates that the amplitude of local CpG density fluctuation varies more drastically along the genome.

### Generalized definition of CGI forest and prairie domains

As many species do not have identified CGIs, we expanded the forest-prairie domains definition (Liu et al, 2018) in human and mouse to the CGI-rich and CGI-poor domains in all species using human and mouse genomes as references. Considering that forest and prairie domains are reflections of the high level of CpG density fluctuation along the sequence, we selected sequential units with significantly high CpG density as CGI-rich domain "loci" at 200 bp, 10, and 500 kb length scales, respectively. We next define regions either with significantly low CpG density (the value of which is smaller than both 75% regions of the species and five percentile of human and mouse forest domains) or with both low CpG density variation (the value of which is smaller than 75% regions of the species and five percentile of human and mouse forest domains) and low CpG density as the CGI-poor "clusters" at 10 kb scale. The CGI-rich and CGI-poor domains are then defined following previous procedures used for forest and prairie domains except that the critical distance is selected as the maximum distance, using which at least 95% CGI-poor "clusters" would be classified into CGI-poor domains. For species with CGI-poor "clusters" taking up <1% of total length or with CGI-rich "loci" taking up more than 60% of total length, we used instead a canonical critical distance (which is the critical neighboring CpG density peak distance defined following our previous work [Liu et al, 2018]). The generalized forest and prairie domains definitions are reflections of the alternation between regions enriched in high CpG density peaks and regions with low CpG densities and small fluctuations.

### Nonnegative matrix factorization

NMF (Lee & Seung, 1999) is a decomposition method for a matrix (i.e., multivariate data) that has been used widely in signal processing, image recognition (Wang et al, 2016; Du & Swamy, 2019) and computational biology (Devarajan, 2008). The aim of nonnegative matrix factorization is to reproduce the observed data by combining a limited number of basis components.

A nonnegative factorization of matrix X is an approximation of X by the product of two matrices W and H, which are constrained to have nonnegative entries (i.e., W ≥ 0, H ≥ 0). This decomposition is thus an approximation, not an equality. The solution W and H matrices minimize the quadratic error between X and W*H. The number of rows of W and columns of H should be same as the ones of X, respectively, and the number of columns of W and rows of H can be chosen as needed.

When it is applied to Hi-C map, each column of W are defined as structure vector for it reflects intensity of Hi-C signal along the DNA sequence. Unlike the compartment vector calculated by PCA, the values of structure vector are all positive (Fig S5 and Table S8). Using structure vectors to recapitulate the Hi-C map is more informative on the contacts between compartment A and B than compartment vectors because there is no offset of positive and negative elements of matrices. In this study, we applied NMF to Observed/Expected Hi-C matrix. The number of columns of W (structure vectors) is set as two, as these two structure vectors were found to be correlated with compartments A and B, respectively.

Besides normal NMF, we also factorized Hi-C matrix by balanced NMF (BNMF) and Graph regularized NMF (GRNMF) to test the robustness of structure vectors (Fig S6).

### Hi-C map analysis

Raw Hi-C data were processed by the ICE procedure. We grouped chromatin loci according to genomic distance, then calculated the decay of average Hi-C contact frequency with the genomic distance. To calculate the structure vectors, the observed/expected Hi-C map was first calculated to ensure the average contact frequency at different genomic distance is normalized, before performing nonnegative matrix factorization. Interaction segregation ratio between A/B compartments is defined as contacts of compartments of the same type (A–A, B–B) divided by contacts of compartments of different types (A-B) of normalized Hi-C contact matrix.

## Data Access

Genomes of species are publicly available in NCBI and UCSC genome browser. Code for CGI-rich/poor domain division, CpG variability calculation, multi-SE analysis, and NMF is available on request.

## Supplementary Information

## Acknowledgements

This work was supported by National Natural Science Foundation of China (92053202, 22050003, 21927901, and 21821004) and the National Key Research and Development Program of China (2017YFA0204702).

### Author Contributions

Z Cai: conceptualization, data curation, formal analysis, and writing—original draft, review, and editing.
Y He: software and methodology.
S Liu: conceptualization, software, and writing—original draft.
Y Xue: data curation and software.
H Quan: data curation.
L Zhang: conceptualization and data curation.
YQ Gao: conceptualization, data curation, supervision, funding acquisition, investigation, methodology, project administration, and writing—original draft, review, and editing.

### Conflict of Interest Statement

The authors declare that they have no conflict of interest.

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
