## [Reviewer comments · Life Science Alliance]

Life Science Alliance

Hierarchical dinucleotide distribution in genome along evolution and its effect on chromatin packing

Zhicheng Cai, Yueying He, Sirui Liu, Yue Xue, Hui Quan, Ling Zhang, and Yi Gao

DOI: <https://doi.org/10.26508/lsa.202101028>

Corresponding author(s): Yi Gao, Peking University

Review Timeline:	Submission Date:	2021-01-21
	Editorial Decision:	2021-03-27
	Revision Received:	2021-05-19
	Editorial Decision:	2021-05-31
	Revision Received:	2021-06-10
	Accepted:	2021-06-11

Transaction Report:

March 27, 2021

Re: Life Science Alliance manuscript #LSA-2021-01028-T

Prof Yi Qin Gao
Peking University
No. 5 Yiheyuan Road
Beijing 100871
China

Dear Dr. Gao,

Thank you for submitting your manuscript entitled "Hierarchical dinucleotide distribution in genome along evolution and its effect on chromatin packing" to Life Science Alliance. The manuscript was assessed by expert reviewers, whose comments are appended to this letter.

We apologize for this extended and unusual delay in getting back to you. As you will note from the reviewers' comments below, the reviewers are interested in these findings, but have also raised several significant concerns, all of which should be addressed prior to further consideration of the manuscript at LSA. We would, thus, like to invite you to submit a revised version of the manuscript that addresses all of the reviewers' points.

Thank you for this interesting contribution to Life Science Alliance. We are looking forward to receiving your revised manuscript.

Sincerely,

Shachi Bhatt, Ph.D.

Executive Editor

Life Science Alliance

<https://www.lsjournal.org/>

Interested in an editorial career? EMBO Solutions is hiring a Scientific Editor to join the international Life Science Alliance team. Find out more here -

https://www.embo.org/documents/jobs/Vacancy_Notice_Scientific_editor_LSA.pdf

B. MANUSCRIPT ORGANIZATION AND FORMATTING:

Reviewer #1 (Comments to the Authors (Required)):

In this manuscript, the authors explore the CpG distribution changes along evolution and its role in chromatin structure and gene expression. They find that, along evolution, the CpG density is decreasing while the variability is increasing, the correlation between CpG distribution with 3D chromatin organization and with gene expression is increasing. Overall, this

paper provides some new insights into the how the 1D genomic sequence influence the 3D structure and gene expression. This work will inspire more researchers to explore the genomic sequence basement for chromatin organization.

Major issues:

The CpG variability assessment

In the first part of this paper, the authors define and calculate the variability of the CpG density based on the whole genome sequence. They find the variability along genome is increasing in higher species. It would be more convincing to evaluate the variability per chromosomes among species. If this increasing variability is a robust evolutionary phenomena, I would expect a general increase among chromosomes of higher species.

The wavelet transform analyses

The authors use the wavelet transform for the CpG density, structure vector and expression levels to explore the similarity among them. It's maybe a little difficult for most biologists to interpret these analyses. It would be helpful to present these results with more intuitive ways. Specifically,

- (1) Figure 5A show the higher correlation between the CpG island and the structure vector in higher species. Maybe with additional presentation like how consistent between CpG-rich/poor domain and A/B compartments and how this consistence change along evolution.
- (2) Figure 5B show the higher correlation between the CpG island and the chromatin organization at the length scale of compartments. Maybe with additional presentation like the respective consistence between CpG-rich/poor domain boundaries with TAD boundaries and with A/B compartments boundaries.
- (3) Figure 6 show the higher correlation between the CpG island and the expression levels in higher species. Maybe with additional presentation like group the genomic regions (genes better) by CpG density, compare the expression levels among groups and compare the correlation among species.

Chromatin structure segregation analysis

In page 16, first paragraph, the chromatin contact frequency decay analyses demonstrate that a more segregated chromatin structure of human and mouse at length short length scale (< 1 Mb). In contrast, more long range contacts (>1 Mb) in other species. Size of A/B compartments is \sim Mb scale and interactions with compartments is also \sim Mb scale. And in the previous presentation (Figures 5A-5C), the results indicate the CpG density is more well correlated with structure vector at compartments scale in human and mouse. These two results seem contradictory. If the authors want to explore whether the chromatin structure is more segregated in higher species, maybe checking the interaction segregation between A/B compartments (interactions of inter-compartments divided by intra-compartments) or CpG-rich/poor domain is a better way.

Structure vectors

In page 13, first paragraph, the authors use nonnegative matrix factorization (NMF) to decompose the Hi-C matrix into two structure vectors. It's quiet smart to choose NMF as it allows only positive coefficients. However, there are several points need to be carefully considered,

- (1) It is unclear what's the structure interpretation of structure vectors. They authors should evaluate the consistence between structure vectors and chromatin structures (like compartments).
- (2) NMF focuses more on local sub-structures compared with PCA. The previous studies use NMF to detect the local interaction domains (Hu et al., 2016; Lee and Roy, 2020). Considering the difference between matrix factorization methods, the authors should evaluate whether the correlation between CpG density and chromatin structure (Figures 4 and 5) is robust to the matrix factorization method.

Minor issues:

Figure 2 color bar legend

It would be helpful if the authors indicate what's meaning of the color bar, column-scaled values?

Spelling/grammar suggestions:

- Page 5 , line 9. 'HiC' should be 'Hi-C'
- Page 8, line 22-24, Page 9, line 17. 'E.coli' should be 'E.coli'
- Page 13, line 3. 'influeced' should be 'influenced'
- Page 13, line 5. 'investigate' should be 'investigate'
- Page 14, line 2. 'compartment' should be 'compartments'
- Page 16, line 7. 'drosphila' should be 'drosophila'
- Page 16, line 19. 'stornng' should be 'strong'
- Page 16, line 20. 'exsits' should be 'exists'
- Page 2, line 7. 'correlates to' should be 'correlate with'

Hu, X., Shi, C.H., and Yip, K.Y. (2016). A novel method for discovering local spatial clusters of genomic regions with functional relationships from DNA contact maps. *Bioinformatics* 32, i111-i120.

Lee, D.-I., and Roy, S. (2020). Graph-regularized matrix factorization for reliable detection of topological units from high-throughput chromosome conformation capture datasets. *bioRxiv*, 2020.2008.2017.254615.

Reviewer #2 (Comments to the Authors (Required)):

The authors analyzed the dinucleotide distribution in 38 publicly available genomes. Their main focus is on the distributions of CpGs since this particular dinucleotide seems to have the largest variability. The authors correlate the CpG distributions with Hi-C data (indicating chromatin organization) for a subset of their species and find a correlation between CpG distribution and chromatin structure. They can show that the CpG distribution patterns are quite different depending on the respective species group. They conclude that CpG density is a better function and structure indicator than the more general G+C content. They also report the interesting finding that the temperature of the living environment seems to have a noticeable effect on the genomic CpG density variability.

Major points:

1) Within the manuscript the authors frequently refer to lower (plants, invertebrates, fish) and higher (mammals, birds) species. They also state "along evolution, the CpG density decreases" or "In early evolution, the CpG density decreases with little change of the mosaicity and in the later stage of evolution, the CpG density remains largely constant but the mosaicity increases."

Such statements indicate a very outdated model of evolution. The terms "lower" and "higher" species should not be used. The classification is anyway very arbitrary since, for example, fish are much more similar to mammals and birds than to the other groups since they are all vertebrates. Separating birds and reptiles seems also unjustified since birds are reptiles. The terms "lower" and

"higher species" should be removed altogether. The authors should just directly state which taxonomic groups they are referring to, e.g. amniotes (reptiles and mammals).

Similar to that the authors should not refer to an early evolution and a later stage of evolution if they are referring to extant species. Early evolution refers to what happened in the past during the evolutionary history of a certain species or group. But not to extant species. There is no direction in evolution, starting from bacteria and leading to mammals. Instead of using "early" and "late" evolution the authors should, again, clearly refer to which taxonomic group they mean.

Minor points:

1) Occasionally, abbreviations are not explained when they are just for the first time, e.g. TAD in the introduction. In general I would recommend to write abbreviations in full again if its used for the first time in a section (introduction, results, ...).

2) The English of the paper is in general quite well. There are some typos throughout the manuscript. It should be therefore be proof-read again.

Some examples:

"and lack of noncoding elements"->"and lacks noncoding elements"

"DNA sequence appear to"->"DNA sequence appears to"

"compatments"->"compartments"

"stornng correlation exsits"->"strong correlation exists"

3) In the introduction the term "CGI(gene)" is used. I'm not sure if the "(gene)" is there by accident. If not, an explanation what the meaning is might be necessary.

Aside from the mentioned major point I think that the data analysis is appropriate and the results are a useful addition to the field.

Therefore, I recommend acceptance with minor revisions. Nevertheless the acceptance should heavily depend on the successful response to the major point.

Dear Dr. Shachi Bhatt:

Thank you for your letter and for the reviewers' comments concerning our manuscript entitled "Hierarchical dinucleotide distribution in genome along evolution and its effect on chromatin packing" (#LSA-2021-01028-T).

Those comments are very helpful for revising and improving our paper. We have studied the comments carefully and have made corrections accordingly. The revisions are marked in the manuscript. Our response to the reviewers' comments is listed in the following:

Reviewer 1:

In this manuscript, the authors explore the CpG distribution changes along evolution and its role in chromatin structure and gene expression. They find that, along evolution, the CpG density is decreasing while the variability is increasing, the correlation between CpG distribution with 3D chromatin organization and with gene expression is increasing. Overall, this paper provides some new insights into the how the 1D genomic sequence influence the 3D structure and gene expression. This work will inspire more researchers to explore the genomic sequence basement for chromatin organization.

Major issues:

The CpG variability assessment

In the first part of this paper, the authors define and calculate the variability of the CpG density based on the whole genome sequence. They find the variability along genome is increasing in higher species. It would be more convincing to evaluate the variability per chromosomes among species. If this increasing variability is a robust evolutionary phenomena, I would expect a general increase among chromosomes of higher species.

Response: We thank the reviewer for these valuable suggestions. We have now added evaluations on the variability for each individual chromosome among different species and added the results in Figure S4. Species can be effectively clustered into different groups by the median value of CpG variabilities. The increasing trend of CpG variability of species can also be observed from this Figure.

The wavelet transform analyses

The authors use the wavelet transform for the CpG density, structure vector and expression levels to explore the similarity among them. It's maybe a little difficult for most biologists to interpret these analyses. It would be helpful to present these results with more intuitive ways. Specifically,

(1) Figure 5A show the higher correlation between the CpG island and the structure vector in higher species. Maybe with additional presentation like how consistent between CpG-rich/poor domain and A/B compartments and how this consistence change along evolution.

Response: We have added the Pearson correlation between CpG-rich/poor domains and A/B compartments of species in Table S9. The Pearson correlation between CpG-rich/poor domains and A/B compartments of species such as human, mouse and chicken is higher than that of *A.thaliana* and zebrafish. We have also showed the comparison of CpG-rich/poor domain index and A/B compartments index of human and *A.thaliana* in Figure S9. We have included related discussions in the text (L273: Consistently, we also compared the correlation between CpG-rich/poor domains and compartments in different species and found that it is higher in human, mouse and chicken than in *A.thaliana* and zebrafish(Table S9).)

(2) *Figure 5B show the higher correlation between the CpG island and the chromatin organization at the length scale of compartments. Maybe with additional presentation like the respective consistence between CpG-rich/poor domain boundaries with TAD boundaries and with A/B compartments boundaries.*

Response: We have added the distance distribution of TAD boundaries and A/B compartments boundaries to CpG-rich/poor domain boundaries in Figure S10. We have also included related discussions in the main text (L284: Consistently, the overlap between A/B compartments and CpG-rich/poor domains is more significant than the one between TADs and CpG-rich/poor domains.)

(3) *Figure 6 show the higher correlation between the CpG island and the expression levels in higher species. Maybe with additional presentation like group the genomic regions (genes better) by CpG density, compare the expression levels among groups and compare the correlation among species.*

Response: We have added comparisons on the expression levels of genes of different CpG densities in human, zebrafish and rice in Figure S13. In human, the expression level of genes generally increases with the CpG density at low CpG densities and then remains largely constant (with a slight decrease) at high CpG densities. The expression level of rice genes is low at both high and low CpG densities. The expression level of genes also shows a maximum at median CpG densities in zebrafish. The expression level of genes thus appears to be more positively correlated with CpG density in human than in zebrafish and in rice.

We have included a related discussion in the revised text (L326: Furthermore, we grouped genes by CpG density and compared the expression levels of genes of different CpG densities. The dependence of gene expression level on CpG density is seen to be different for different species (Figure S13). For example, the expression levels of zebrafish and rice genes show a more pronounced peak at median CpG densities than that of human.)

Chromatin structure segregation analysis

In page 16, first paragraph, the chromatin contact frequency decay analyses demonstrate that a more segregated chromatin structure of human and mouse at length short length scale (< 1 Mb). In contrast, more long range contacts (>1 Mb) in

other species. Size of A/B compartments is ~Mb scale and interactions with compartments is also ~Mb scale. And in the previous presentation (Figures 5A-5C), the results indicate the CpG density is more well correlated with structure vector at compartments scale in human and mouse. These two results seem contradictory. If the authors want to explore whether the chromatin structure is more segregated in higher species, maybe checking the interaction segregation between A/B compartments (interactions of inter-compartments divided by intra-compartments) or CpG-rich/poor domain is a better way.

Response:

Chromatin structure of human and mouse shows a strong segregation at a length scale similar to the length of one compartment. Strong contacts at the sub-Mb scale in human and mouse chromatin corresponds to a their well-resolved compartment formation. On the other hand, for human and mouse, the CpG density correlates well with structure vector, indicating fluctuations of CpG density and structure vector along the genome resemble each other at the compartments scale. Such a result indicates the significant contact difference between different CpG density regions, i.e. preferred interactions are formed between compartments of the same type (A-A or B-B) over compartments of different types(A-B). The two results mentioned by the reviewer thus reflect intensity of contacts at intra- and inter-compartment level respectively. We thank the reviewer for pointing out this possible confusion in the previous presentation.

Following the reviewer's suggestion, we have added an analysis on the interaction segregation ratio between A/B compartments (interactions of compartments of the same type divided by compartments of different types) in Table S10. It can be seen that the interaction segregation ratio of human and mouse is higher than other species.

We have also included a related discussion (L314: we also calculated the interaction segregation ratio of compartments A/B for different species (Table S10). The interaction segregation ratio of human and mouse is higher than other species.)

Structure vectors

In page 13, first paragraph, the authors use nonnegative matrix factorization (NMF) to decompose the Hi-C matrix into two structure vectors. It's quite smart to choose NMF as it allows only positive coefficients. However, there are several points need to be carefully considered,

(1) It is unclear what's the structure interpretation of structure vectors. They authors should evaluate the consistence between structure vectors and chromatin structures (like compartments).

Response: We calculated the Pearson correlation of structure vectors and compartment vector. We found that one of the structure vectors is positively correlated with compartment vector, and the other is negatively correlated with compartment vector. We have listed the max absolute value of Pearson correlation of two structure vectors with compartment vector of species in Table S8. We also showed compartment and structure vectors of chicken in Figure S5, which can be seen to be

closely correlated.

(2) *NMF focuses more on local sub-structures compared with PCA. The previous studies use NMF to detect the local interaction domains (Hu et al., 2016; Lee and Roy, 2020). Considering the difference between matrix factorization methods, the authors should evaluate whether the correlation between CpG density and chromatin structure (Figures 4 and 5) is robust to the matrix factorization method.*

Response: The two references mentioned by the reviewer applied balanced nonnegative matrix factorization (BNMF) and Graph regularized nonnegative matrix factorization (GRNMF) in their studies, respectively. As pointed out by the reviewer, these methods were used to interrogate the local chromatin structures such as TADs.

We have added comparison of structure vectors factorized by NMF, BNMF and GRNMF in chicken, zebrafish, *A.thaliana*, with the results given in Figure S6. Structure vectors factorized by different matrix factorization methods are very similar to each other in trend, showing that the calculation on structure vector is robust to the matrix factorization methods.

We have also included related discussions (L258: We tested the robustness of structure vectors and found structure vectors factorized by different Nonnegative Matrix Factorization methods to be similar to each other after scaling (Figure S6).).

Minor issues:

Figure 2 color bar legend

It would be helpful if the authors indicate what's meaning of the color bar, column-scaled values?

Response: The color bar shows column-scaled values. We added the description for the color bar in the Figure 2 legend.

Spelling/grammar suggestions:

- Page 5 , line 9. 'HiC' should be 'Hi-C'
- Page 8, line 22-24, Page 9, line 17. 'E.coli' should be 'E.coli'
- Page 13, line 3. 'influeced' should be 'influenced'
- Page 13, line 5. 'invesitgate' should be 'investigate'
- Page 14, line 2. 'compatment' should be 'compartments'
- Page 16, line 7. 'drosphila' should be 'drosophila'
- Page 16, line 19. 'stornng' should be 'strong'
- Page 16, line 20. 'exsits' should be 'exists'
- Page 2, line 7. 'correlates to' should be 'correlate with'

Response: We thank the reviewer for these corrections. We have corrected these mistakes in the manuscript.

Reviewer 2:

The authors analyzed the dinucleotide distribution in 38 publicly available genomes. Their main focus is on the distributions of CpGs since this particular dinucleotide seems to have the largest variability. The authors correlate the CpG distributions with Hi-C data (indicating chromatin organization) for a subset of their species and find a correlation between CpG distribution and chromatin structure. They can show that the CpG distribution patterns are quite different depending on the respective species group. They conclude that CpG density is a better function and structure indicator than the more general G+C content. They also report the interesting finding that the temperature of the living environment seems to have a noticeable effect on the genomic CpG density variability.

Major points:

1) Within the manuscript the authors frequently refer to lower (plants, invertebrates, fish) and higher (mammals, birds) species. They also state "along evolution, the CpG density decreases" or "In early evolution, the CpG density decreases with little change of the mosaicity and in the later stage of evolution, the CpG density remains largely constant but the mosaicity increases."

Such statements indicate a very outdated model of evolution. The terms "lower" and "higher" species should not be used. The classification is anyway very arbitrary since, for example, fish are much more similar to mammals and birds than to the other groups since they are all vertebrates. Separating birds and reptiles seems also unjustified since birds are reptiles. The terms "lower" and "higher species" should be removed altogether. The authors should just directly state which taxonomic groups they are referring to, e.g. amniotes (reptiles and mammals).

Similar to that the authors should not refer to an early evolution and a later stage of evolution if they are referring to extant species. Early evolution refers to what happened in the past during the evolutionary history of a certain species or group. But not to extant species. There is no direction in evolution, starting from bacteria and leading to mammals. Instead of using "early" and "late" evolution the authors should, again, clearly refer to which taxonomic group they mean.

Response: We thank the reviewer for these good suggestions. We have modified the corresponding statements in the manuscript.

(For example, L229: Different from CpG density, the CpG variabilities of invertebrates and plants are all generally very similar and small. In contrast, the variabilities of mammals and birds are much higher than those of other species.

L232: During the evolution of invertebrates and plants, the CpG density decreases with little change of the genome mosaicity and during the evolution of mammals and birds, the CpG density remains largely constant but the genome mosaicity increases.

L390: The sequence properties of bacteria, plants, invertebrates, fishes and reptiles are distinctly different from birds and mammals in terms of the dinucleotide distribution.)

Minor points:

1) Occasionally, abbreviations are not explained when they are just for the first time, e.g. TAD in the introduction. In general I would recommend to write abbreviations in full again if its used for the first time in a section (introduction, results, ...).

Response: We thank the reviewer for these valuable suggestions. We have made changes following the reviewer's suggestion and added the full description.

2) The English of the paper is in general quite well. There are some typos throughout the manuscript. It should be therefore be proof-read again.

Some examples:

"and lack of noncoding elements"->"and lacks noncoding elements"

"DNA sequence appear to"->"DNA sequence appears to"

"compatments"->"compartments"

"stornng correlation exsits"->"strong correlation exists"

Response: We thank the reviewer for these corrections. We have corrected these typos in the manuscript.

3) In the introduction the term "CGI(gene)" is used. I'm not sure if the "(gene)" is there by accident. If not, an explanation what the meaning is might be necessary.

Response: CGI forest domains are generally the regions with high gene density and high CGI density, CGI prairie domains are the regions with low gene density and depleted for CGI. We added the 'gene' after 'CGI' to emphasize the coexistence of gene and CGI on the genome.

We have added the explanation in the text (L76: the distribution of which is also heterogeneous and correlated with gene density on the genome.)

Sincerely yours,

Yi Qin Gao and Zhicheng Cai.

May 31, 2021

RE: Life Science Alliance Manuscript #LSA-2021-01028-TR

Prof. Yi Qin Gao
Peking University
Beijing National Laboratory for Molecular Sciences, College of Chemistry and Molecular Engineering
No. 5 Yiheyuan Road
Beijing 100871
China

Dear Dr. Gao,

Thank you for submitting your revised manuscript entitled "Hierarchical dinucleotide distribution in genome along evolution and its effect on chromatin packing". We would be happy to publish your paper in Life Science Alliance pending final revisions necessary to meet our formatting guidelines.

Please also attend to the following,

- please upload both your main and supplementary figures as single files, not as part of the main text
- please add your main, supplementary figure, and table legends to the main manuscript text after the references section;
- please add ORCID ID for the corresponding author-you should have received instructions on how to do so
- we encourage you to revise the figure legends for figures S11 such that the figure panels are introduced in an alphabetical order
- please be sure to add callouts for all main and supplementary figures (including panels) to your main manuscript text. For example, please be sure to add callouts for S1A-D or Figure 3A, B...etc.
- please add the methods from the supplementary text document to the main manuscript, the supplementary tables and supplementary figures as separate files to the system, and the legends for supplementary tables and figures to the main manuscript

A. FINAL FILES:

B. MANUSCRIPT ORGANIZATION AND FORMATTING:

Sincerely,

Shachi Bhatt, Ph.D.
Executive Editor

Life Science Alliance
<http://www.lsjournal.org>
Tweet @SciBhatt @LSAJournal

Reviewer #1 (Comments to the Authors (Required)):

Cai et al return a revised manuscript, in which they perform new analyses to confirm and refine their previous analyses. The new information provides reasonable responses to the concerns raised in the initial manuscript. I recommend the manuscript for publication in LSA.

June 11, 2021

RE: Life Science Alliance Manuscript #LSA-2021-01028-TRR

Prof. Yi Qin Gao
Peking University
Beijing National Laboratory for Molecular Sciences, College of Chemistry and Molecular Engineering
No. 5 Yiheyuan Road
Beijing 100871
China

Dear Dr. Gao,

Thank you for submitting your Research Article entitled "Hierarchical dinucleotide distribution in genome along evolution and its effect on chromatin packing". It is a pleasure to let you know that your manuscript is now accepted for publication in Life Science Alliance. Congratulations on this interesting work.

DISTRIBUTION OF MATERIALS:

Again, congratulations on a very nice paper. I hope you found the review process to be constructive and are pleased with how the manuscript was handled editorially. We look forward to future exciting submissions from your lab.

Sincerely,
